# Establishment and Characterization of a Spermatogonial Stem Cell Line from Tiger Puffer Fish (*Takifugu rubripes*)

**DOI:** 10.3390/ani13182959

**Published:** 2023-09-19

**Authors:** Leilei Tan, Qian Liu, Yangbin He, Jingjing Zhang, Jilun Hou, Yuqin Ren, Wenxiu Ma, Qian Wang, Changwei Shao

**Affiliations:** 1Jiangsu Key Laboratory of Marine Biological Resources and Environment/Jiangsu Key Laboratory of Marine Biotechnology, Jiangsu Ocean University, Lianyungang 222000, China; tanll2023@163.com; 2National Key Laboratory of Mariculture Biobreeding and Sustainable Goods, Yellow Sea Fisheries Research Institute, Chinese Academy of Fishery Sciences, Qingdao 266071, China; liuqian97927@163.com (Q.L.); heyb1002@163.com (Y.H.); jingjingzhang0307@163.com (J.Z.); mawenxiu121@163.com (W.M.); 3Hebei Key Laboratory of the Bohai Sea Fish Germplasm Resources Conservation and Utilization, Beidaihe Central Experiment Station, Chinese Academy of Fishery Sciences, Qinhuangdao 066100, China; houjl@bces.ac.cn (J.H.); renyq@bces.ac.cn (Y.R.); 4Laboratory for Marine Fisheries Science and Food Production Processes, Qingdao National Laboratory for Marine Science and Technology, Qingdao 266237, China

**Keywords:** spermatogonial stem cell line, cryopreservation, transfection, *Takifugu rubripes*

## Abstract

**Simple Summary:**

Tiger puffer fish (*Takifugu rubripes*) is an economically important marine fish, in which the development of selective breeding is limited due to the long time of sexual maturation. In order to promote the breeding research of this species, we established the spermatogenic stem cell line of the tiger puffer fish. In this study, we isolated spermatogonial stem cells from the testis of the 6-month-old tiger puffer fish and cultured them stably in vitro for more than 60 generations. The characteristics of the in vitro cultured spermatogonial stem cell line showed polygonal in shape. The germ cell marker genes *dnd*, *ddx4*, *piwil*, *dazl*, stem cell marker genes *sox2*, *myca* and *nanog*, and spermatogonial stem cell marker genes *gfra1b* and *ly75* were highly expressed in this cell line. The establishment of this spermatogonial stem cell line provides a dependable material for the reproduction research of tiger puffer fish.

**Abstract:**

Tiger puffer fish (*Takifugu rubripes*) has become the main fish species cultured in China since the last century because of its high economic value. Male and female tiger puffer fish need 2 and 3 years each to reach sexual maturity, which limits the development of breeding research for this species. In recent years, in vitro culture of fish spermatogonial stem cells (SSCs) have shown potential in aquaculture. In the present study, we established a spermatogenic stem cell line from *T. rubripes* (TrSSCs). TrSSCs were characterized by polygonal morphology, predominantly retained 44 chromosomes, and grew rapidly at 26 °C and in L-15. TrSSCs were still able to grow stably after more than one year of in vitro culture. TrSSCs showed positive alkaline phosphatase staining. TrSSCs expressed germ cell-associated genes, including *dnd*, *ddx4*, *piwil*, *gfra1b*, *sox2*, *myca*, *nanog*, *ly75*, and *dazl*, as determined by semiquantitative assays, and almost all cells were found to express the germ cell genes *ddx4* and *gfra1b* in a fluorescence in situ hybridization assay. In vitro, induction experiments demonstrated the TrSSCs possessed the ability to differentiate into other types of cells. Our research has enriched the fish spermatogonial stem cell resource bank, which will provide an efficient research model for sex determination and sex control breeding in fish, establishing a foundation for subsequent breeding research.

## 1. Introduction

Spermatogonial stem cells (SSCs) can self-renew to produce more stem cells and produce spermatozoa through the differentiation of daughter cells and further differentiation [1,2,3]. SSCs represent only a small proportion of testicular cells [4,5], and the establishment of an SSC line can generate more SSCs in vitro, which is important for breeding research. Mammalian SSC lines have been established in many species, including mouse (*Mus musculus*), rat (*Rattus norvegicus*), wild boar (*Sus scrofa*), and tree shrew (*Tupaia belangeri*). These SSC lines can stably proliferate for multiple generations in vitro for a long time and express SSC marker genes [6,7,8,9,10].

In fish, SSC establishment is limited by cultivation techniques for prolonged growth of spermatogonial stem cells in vitro [11]. Rainbow trout (*Oncorhynchus mykiss*) type A spermatogonia can be cultured in vitro for approximately 1 month, but the growth of SSCs is inhibited by the dominance of testicular somatic cells with increasing incubation time [12,13]. Swamp eel (*Monopterus albus*) SSCs were cultured in vitro for only 1 month without feeder layer cells [14]. Caspian trout (*Salmo caspius*) SSCs still express SSC-related genes (*ly75*, *gfrα1*, *nanos2*, *plzf*, and *vasa*) when cultured in vitro for more than 1 month [15]. To date, three species of fish SSC lines have been cultured in vitro for more than 50 generations. The medaka (*Oryzias latipes*) SSC line maintains the phenotype and gene expression profile (*vasa*, *dazl*, *piwi*, and *c-kit*) of spermatogonial stem cells after 140 passages [16]. Chinese hook snout carp (*Opsariichthys bidens*) SSCs can grow stably after two years of in vitro culture and have SSC gene expression patterns (*dazl*, *dnd*, *vasa*, *gfra1*, and *nanog*) [17]. Orange-spotted grouper (*Epinephelus coioides*) SSCs can proliferate stably in vitro for more than 20 months and express SSC-specific genes (*vasa*, *dazl*, *plzf*, *nanog*, *oct4*, and *ssea1*) [11].

Tiger puffer fish (*Takifugu rubripes*) belongs to the order Tetraodontidae, suborder Tetraodontidae, and family *Tetraodontidae*, is mainly found in coastal China, the Korean Peninsula, and Japan, and is popular for its tasty meat and high nutritional value [18,19]. Since the last century, the tiger puffer fish has gradually become a major aquaculture species in China because of its high economic value [20]. The mature testes of male tiger puffer fish are delicious, so males are more expensive than females in the market [21,22]. Tiger puffer fish grow slowly, taking 17–18 months to reach a weight of 1 kg [23]. It will take 2 and 3 years to reach sexually mature for males and females, respectively, and the size of the parents is larger than the average size on the market. The parental breeding of tiger puffer fish requires more time and space than that of market-sized fish [24,25], which poses a barrier to breeding. Thus, by establishing a SSC line, breeding research can be conducted using stem cells and a new model for SSC research on fish will be established.

In this study, we identified and established a *T. rubripes* SSC line (TrSSCs) which is capable of long-term in vitro cultivation. Subsequently, we investigated the chromosome number, transfection efficiency, and optimal culture conditions and analyzed the gene expression pattern of TrSSCs. This study provides a new model for SSC research and an unlimited source of cells.

## 2. Materials and Methods

### 2.1. Fish and Sample Preparation

Six-month-old tiger puffer fish was provided by Beidaihe Central Experiment Station, Chinese Academy of Fishery Sciences. They were reared in circular tanks placed in 300 L of seawater at 23 °C under a 14 h/10 h light-dark photoperiod. The fish were fed pellet feed and seawater was changed twice a day.

Blood was collected from healthy Japanese flounder (*Paralichthys olivaceus*) using a sterile syringe and dispensed in centrifuge tubes. The dispensed blood was left to stand for 2 h at 4 °C, and the rest of the blood was processed by centrifugation at 8000 rpm for 15 min at 4 °C to collect serum. Sterile serum was collected using a 0.22 μm sterile filter. The culture medium contained Leibovitz’s L-15 (L-15, Gibco, Billings, MT, USA), 15% fetal bovine serum (FBS, Gibco, Billings, MT, USA), 800 IU/mL penicillin, 0.8 mg/mL streptomycin and 2 μg/mL amphotericin B (Solarbio, Beijing, China), 1% Japanese flounder serum, 1% Japanese flounder embryo extract, 2 ng/mL recombinant human fibroblast growth factor (bFGF, Abcam, Cambridge, UK) and 2 ng/mL recombinant human leukemia inhibitory factor (LIF, Abcam, Cambridge, UK).

### 2.2. Primary Culture and Subculture

Healthy fish were chosen and raised in sterile seawater containing 1000 IU/L penicillin and 1000 mg/L streptomycin for one day. Then they were placed into a water body containing 200 mg/L MS-222 anesthetic followed by wiping with a 75% alcohol cotton ball. Subsequently, the testes were removed on an ultraclean bench, washed three times with L-15 containing penicillin-streptomycin-amphotericin B to remove the tissue mesentery and blood, and cut into 1 mm^3^ small pieces. These pieces were digested with 0.05% trypsin–EDTA for 30 min and inoculated into a 25 cm^2^ culture flask for culture at 23 °C. During the primary cell culture, half of the medium was changed every 3 days. The cell morphology was observed under a microscope, and when the cells reached about 50% confluence, the area of higher density of fibroblasts was marked on the top of the culture flask, and then the fibroblasts were removed with a cell scraper.

After the tiger puffer testis primary cells had reached 90% confluence, the cells were digested with 0.25% trypsin to remove small pieces of the attached testis. The SSCs were first digested under trypsin treatment for a shorter period of time, due to the relatively weak adsorption properties of SSCs to the culture flasks [26], and the SSCs to be isolated were treated with this method for the first 10 generations.

When the cells adhered and grew to 80–90% of the bottom area of the culture flask, the passage was carried out. After discarding the old medium, cells were washed with L-15 containing penicillin–streptomycin-amphotericin B. Approximately 1 mL of trypsin–EDTA was added, and the cell culture flask was gently shaken in order to spread the trypsin evenly. After most cells became round, trypsin was removed, the cells were subcultured at a ratio of 1:2, a new culture medium was added, the cells were gently blown several times to disperse and then cultured in a 23 °C incubator.

### 2.3. Cryopreservation and Recovery of Cells

TrSSCs in good condition were collected with 0.25% trypsin when they reached 90% fusion and were centrifuged at 1500 rpm for 5 min at 23 °C in order to discard the trypsin. The pelleted cells were collected in a cell culture freezing medium (70% L-15 medium, 20% FBS, 10% DMSO) to resuspend the cells in 2 mL cryogenic vials. The cryogenic vials were transferred into a programmed cryopreservation container (Beyotime, Jiangsu, China) for storage at −80 °C for 12 h and then kept in liquid nitrogen (−196 °C) for long-term storage.

For reculture, the cryopreserved cells were shaken in a 37 °C water bath until melted. The cells were then added to 2 mL centrifuge tubes at 1500 rpm for 5 min at room temperature, collected in fresh subculture medium, seeded into 25 cm^2^ cell culture flasks, and incubated at 23 °C.

### 2.4. Cell Growth Studies

The growth status of cells was detected in the 30th generation at different temperatures (17 °C, 20 °C, 23 °C, 26 °C and 29 °C) and media (DMEM, DMEM/F12, M199, MEM and L-15). The 30th generation cells were made into a cell suspension of appropriate concentration calculated with the blood cell counting plate. Then, the cell suspension was added to the 96-well plate at 100 μL/well (approximately 2000 cells per well, 5 replicates). The culture plate was placed in the incubator for preculture for 12 h, the experimental temperature or test medium was changed after 12 h, and the cells were cultured in the incubator for an appropriate time. The number of cells in each well of the culture plate was detected daily for 7 consecutive days using the CCK-8 kit (Yeasen, Shanghai, China) according to the manufacturer’s instructions.

### 2.5. Chromosome Karyotype Analysis

Details of chromosome karyotype analysis were as previously described [27], with modifications. TrSSCs at passage 40 were inoculated into 75 cm^2^ culture flasks for 24 h at 23 °C. The incubated cells were stimulated for 30 min at 4 °C, transferred to the culture medium and cultured in an incubator for 16 h at 23 °C. Colchicine was added to the cells at a final concentration of 0.05 μg/mL and then incubated for 4 h. After removal of the original medium PBS (Solarbio, Beijing, China) rinse, cells were collected by trypsin treatment. The collected cells were treated in hypotonic solution (0.075 M KCl) for 40 min at 37 °C and centrifuged at 1500 rpm for 5 min. Then, the supernatant was removed and precooled Karnaugh’s solution was added and to fix cells for 15 min at 4 °C. The tubes were centrifuged as described above and the cell pellets were dissolved in the fixative and dropped onto slides. After drying, the slides were stained with 5% Giemsa dye solution (Solarbio, Beijing, China) for 25 min. The slides were washed and dried, and then observed and photographed under a microscope.

### 2.6. Transfection with GFP Reporter Gene

The efficiency of cell transfection was examined by transfecting plasmid pEGFP-N1 expressing green fluorescent protein. Cells at passage 30 were inoculated into 12-well plates and stably grown for 1 day before transfection until cell confluence reached 80% for cell transfection. Before transfection, the old culture medium was discarded, the cells were washed once with PBS, and 1 mL of L-15 containing 3% FBS was added to each well. According to the manufacturer’s instructions, Lipofectamine^TM^ 3000 and P3000^TM^ (Invitrogen, Waltham, MA, USA) were diluted by Opti-MEM (Gibco, Billings, MT, USA) and mixed with pEGFP-N1. The mixture of pEGFP-N1 and liposomes was incubated for 15 min, added to the cell culture plate, and cultured at 23 °C. After 2 days, the cells were observed under a fluorescence microscope (DMi8, Leica, Wetzlar, Germany). The transfection efficiency of the cell lines was calculated using the software ImageJ (version 1.53).

### 2.7. Alkaline Phosphatase Staining

TrSSCs at passage 30 were incubated in 6-well plates until approximately 60% confluence by microscopic observation, and then were fixed in 4% paraformaldehyde for 15 min and washed 3 times with PBS. TrSSCs were stained for alkaline phosphatase for 24 h by using the BCIP/NBT alkaline phosphatase color development kit (Beyotime, Jiangsu, China) according to the manufacturer’s instructions. After 24 h, the BCIP/NBT dye working solution was removed. Cells were washed twice with distilled water to stop the color reaction, and transferred to an inverted microscope (Leica, Wetzlar, Germany) for observation.

### 2.8. Fluorescence In Situ Hybridization

TrSSCs at a density of 5 × 10^4^ were inoculated into six-well plates with slides placed on the bottom (NEST, Wuxi, China) until cell fusion reached 60% and fixed in 4% paraformaldehyde for 20 min. Fluorescence in situ hybridization probes were designed according to the instructions for the synthesis of digoxin fluorescent labeling (Servicebio, Wuhan, China). Fluorescence in situ hybridization was performed with a FISH kit (Gefan, Shanghai, China) according to the manufacturer’s requirements. Images were acquired and analyzed using a laser confocal microscope. The probe sequences were shown in Table 1.

### 2.9. Total DNA and RNA Extraction and Polymerase Chain Reaction

Total DNA was extracted from 30th generation cells, testes, and ovaries using a DNA Extraction Kit (TianGen, Beijing, China) according to the manufacturer’s instructions. The total DNA quantity and quality were detected with a microspectrophotometer.

Total RNA from cells, testes, and ovaries was extracted using TRIzol (Invitrogen, Waltham, MA, USA). RNA was extracted in chloroform, precipitated in isopropanol, washed twice with 75% alcohol, and then dissolved in DEPC water. The quality of RNA was detected by agarose gel electrophoresis. The cDNA was synthesized using the PrimeScript™ RT Reagent Kit with gDNA Eraser (TaKaRa, Kusatsu, Japan) from 1.0 µg total RNA samples following the manufacturer’s manual.

Semiquantitative PCR was used to detect the origin of the cells and the expression of cellular marker genes in cells and gonads. The PCR system was as follows: upstream and downstream primers (10 μmol/L) 0.5 μL each; DNA template (90 ng/μL) 1 μL; RNA-free water 8 μL; 2 × EasyTaq SuperMix 10 μL (Vazyme, Nanjing, China). The PCR conditions were as follows: 94 °C predenaturation for 10 min; 35 cycles: denaturation at 94 °C for 30 s, renaturation at 55 °C for 30 s, and extension at 72 °C for 45 s; extension at 72 °C for 10 min; and storage at 4 °C. PCR products were analyzed by 1% agarose gel electrophoresis for specific bands. Primers were shown in Table 1.

### 2.10. Induced Differentiation of TrSSC In Vitro

TrSSCs at passage 30 were cultured in suspension in 6-well plates for 7 days in a culture containing 10 μM retinoic acid. After 7 days of culture, the cells were isolated and inoculated into 12-well plates for overnight cell adhesion. Then cells were observed and photographed under a microscope.

## 3. Results

### 3.1. Establishment of a Spermatogonial Stem Cell Line

Primary cells migrated from the testis tissue mass after 5 days of primary culture at 23 °C (Figure 1a) and formed a monolayer of cells at 20 days at 23 °C (Figure 1b). Cells in the initial culture were somatic cells and germ cells with distinct nuclei. As the transmission time increased, a gradual decrease in the number of somatic cells with fibroblast-like phenotype could be observed, with cells gradually becoming predominantly polygonal spermatogonial stem cells (Figure 1c,d). The subcultured cells were passaged every 3–5 days for one to two passages. Spermatogonial stem cells were obtained from male *T. rubripes*, stably passaged for more than 60 generations and named *T. rubripes* spermatogonial stem cells (TrSSCs).

In addition, different generations of cells were frozen in liquid nitrogen for seed preservation. Frozen cells were rapidly thawed and further cultured and no changes in morphology and growth were detected. The cells at passage 30 were observed using an inverted microscope to have a high rate of attachment at 5 d of recovery and to reach 90% confluence (Figure 1e,f).

### 3.2. Effects of Temperature and Medium on Cell Growth

TrSSCs at passage 30 were used to evaluate the effect of culture temperature and medium on the cell growth rate. The results indicated that TrSSCs grew stably in different media containing 15% FBS, and the growth rate was fastest in the L-15 medium at 23 °C. TrSSCs grown in MEM, DMEM/F12, and M199 grew stably without significant differences (Figure 2a). TrSSCs can survive and grow in an L-15 medium containing 15% FBS at 17–29 °C, with better growth rates at 26 °C and 29 °C (Figure 2b).

### 3.3. GFP Reporter Transfection

To detect the gene function and genetic manipulation research ability of the TrSSCs, we transfected 30th generation cells with plasmid pEGFP-N1. As shown in Figure 3, fluorescence microscopy results showed that green fluorescent signals were observed in cells transfected for 24 h, and the transfection efficiency was calculated at 36% by using the software ImageJ (version 1.53), indicating that pEGFP-N1 was transfected into cells, and the cell line can be used to study the expression and function of foreign genes in vitro.

### 3.4. Chromosome Numbe

For chromosome karyotype analysis, 100 cells (TrSSCs at passage 40) were observed and counted. The chromosome morphology of TrSSC diploids was shown in Figure 4a, of which 62 TrSSCs had a diploid karyotype of 44 chromosomes (Figure 4b).

### 3.5. Characterization of the Spermatogonial Property of T. rubripes TrSSCs

The tiger puffer fish *cox1* partial gene was amplified from the cDNA of the TrSSCs to verify the species of the cell line. The results of the verification of cell line origin and type were shown in Figure 5a. The PCR product was 883 bp in length and showed a 100% sequence similarity with the sequences of *cox1* from tiger puffer fish (Genbank accession number: AJ421455) (Appendix A). Amplification of the *cox1* gene in the cell line was consistent with that in the ovary and testis (Figure 5a). These results indicated that the cell line was derived from tiger puffer fish.

In our study, when testicular cells from *T. rubripes* were stably subcultured for 30 passages, 60% of adherent cells were found to be strongly positive for alkaline phosphatase staining (Figure 5b). The identification of TrSSCs was then analyzed by using germ cell markers. The expression of stem cell-related marker genes *sox2* and *myca*, germ cell-specific marker genes *dnd* and *ddx4*, and the reproductive stem cell marker genes *piwil*, *gfra1b*, *dazl*, *nanog*, and *ly75* were detected (Figure 5a). We also examined the expression of germ cell-specific marker genes and reproductive stem cell marker genes by fluorescence in situ hybridization. Almost all cells were positive for the markers *ddx4* (Figure 5c–e). More importantly, almost all the cells expressed *gfra1b* (Figure 5f–h). These results indicated the successful culture of testicular cells.

### 3.6. Induced differentiation of TrSSC In Vitro

After continuous treatment with retinoic acid for one week, the types of terminally differentiated cells from TrSSCs were identified, including fibroblasts (Figure 6a) and astrocytes (Figure 6b), suggesting the pluripotency ability of TrSSCs in vitro.

## 4. Discussion

In this study, we used an effective method to establish a spermatogenic stem cell line (TrSSCs) from the testis of a six-month-old tiger puffer that has been continuously passaged for more than 60 generations. The optimal conditions for the TrSSCs growth rate were determined to be at L-15 and at 26 °C. TrSSCs were validated as spermatogonial stem cell lines through three biological strategies: strong staining activity of alkaline phosphatase, expression of SSC-related genes, and in vitro differentiation analysis.

Genetic material is carried by chromosomes, and the number of chromosomes is determined for each species. TrSSCs have 44 chromosomes, the same number of chromosomes as in the kidney tissue of tiger puffer fish (2n = 44) [28,29] and the same number of chromosomes as the two-spotted puffer ovary and testis cell line [30,31].

As previously shown, the in vitro cultured SSCs were round and polygonal in shape, positive for alkaline phosphatase activity, and expressed germ cell genes [11]. TrSSCs originate from the gonads, can stably pass through generations, and maintain a polygonal cell morphology. Alkaline phosphatase is expressed at high levels in embryonic germ cells, embryonic stem cells, and other pluripotent stem cell types [32]. Alkaline phosphatase is abundant in the spermatogonial stem cells of medaka, orange-spotted grouper (*Epinephelus coioides*) [11], and Chinese hook snout carp [17]. TrSSCs are positive for alkaline phosphatase and express a variety of spermatogonial stem cell-related genes, including *dnd*, *ddx4*, *piwil*, *nanog*, *ly75*, *gfra1b*, *dazl*, *sox2*, and *myca*. *Sox2* plays a role in the maintenance of germ cell proliferation in fish and is present in the spermatogonial stem cells of rohu carp [33]. *myca* is a marker gene associated with cell stem proliferation and is expressed in zebrafish embryos [34]. *dnd* [35] was a marker gene of type A spermatogonia in gibel carp [36] and Pacific bluefin tuna [37]. *ddx4* was highly expressed in zebrafish spermatogonial stem cells [38]. *piwil* showed expression in both turbot [39] and zebrafish spermatogonia [40]. *nanog* was a marker of fish pluripotent stem cells [41] and was expressed in medaka [42] and Japanese flounder spermatogonia [43,44]. *dazl* was shown to be highly expressed in spermatogonia of medaka [45] and *Coilia nasus* [46]. *ly75* was a marker gene of turbot spermatogonial stem cells [47] and was also specifically expressed in rainbow trout type A spermatogonia [48]. *gfra1* is a highly conserved molecular marker in vertebrates, and these molecules are considered classical markers of rodent spermatogonial stem cells, which were key regulators in maintaining and regulating mammalian SSC self-renewal [49,50,51]. *gfra1* was shown to be expressed in spermatogonial stem cells of Nile tilapia [52], dogfish [53], rainbow trout [54], and swamp eel [14]. Medaka SSCs express *vasa*, *dazl*, *piwi*, and *c-kit* [16]. Two-year in vitro stable culture of the Chinese hook snout carp SSC line showed *dazl*, *vasa*, *dnd*, *gfra1*, and *nanog* expression and weak expression of *dmrt1* [17]. Spotted grouper SSC lines are alkaline phosphatase-positive and express the germ cell markers *vasa*, *dazl*, and *plzf* and the stem cell markers *nanog*, *oct4*, and *ssea1* [11]. In short, TrSSCs can stably self-renew in vitro and exhibit the properties of spermatogonial stem cells.

In our study, the culture medium was an important factor for the long-term culture of SSCs in vitro. In rainbow trout, inhibition of A-type spermatogonia growth due to in vitro culture conditions containing high concentrations of FBS which leads to somatic cell overgrowth [12]. The concentration of FBS in the TrSSC culture set was 15%, the same concentration as in the Chinese hook snout carp SSC line [17] and orange-spotted grouper SSC line [11]. In orange-spotted grouper, the bFGF and LIF can play an important role in maintaining the properties of orange-spotted grouper SSCs [11]. Mammalian recombinant GDNF and LIF have no effect on the growth of rainbow trout A-type spermatogonia by in vitro expansion [55]. The addition of rainbow trout serum to the culture medium increases the number and mitotic activity of spermatogonia cultured in vitro [13]. In medaka, bFGF inhibits spermatogonia differentiation and promotes proliferation in vitro [16], and the addition of baculovirus-produced LIF to the culture medium induces spermatogonia proliferation [56]. Fish serum and embryo extracts play an important role in the in vitro self-renewal of spermatogonial stem cells in medaka, zebrafish, Chinese hook snout carp, and Nile tilapia [16,17,45,57]. We used a variety of nutrients and cytokines to maintain TrSSCs in vitro for long-term development, including bFGF, LIF, as well as serum and embryo extracts from Japanese flounder. Although Japanese flounder and tiger puffer belong to different orders, these reagents are important for maintaining the self-renewal of TrSSCs in vitro.

In our study, TrSSC was suitable for culture in L-15, although the media for Chinese hook snout carp and spotted grouper were Dulbecco’s modified Eagle medium [11,17]. The TrSSC culture environment contains sufficient oxygen. In fish, spermatogonial stem cells can be cultured in vitro under saturated oxygen for long periods of time, for example, spermatogonial stem cells of Chinese hook snout carp and spotted grouper can survive under aerobic culture conditions for long periods of time and more than one year [11,17].

Spermatogonial stem cells have the potential to differentiate multiple cell types after induction [58]. Human spermatogonial stem cells can differentiate into cells with neuronal functions in vitro [59]. Spermatogonial stem cells of the Chinese hook snout carp can produce different types of somatic cells in vitro when stimulated by retinoic acid [17]. In our study, TrSSCs could generate fibroblasts and astrocytes in vitro, indicating their capacity for differentiation in vitro.

## 5. Conclusions

In conclusion, SSCs is a special type of cells that capable of self-renewal, exists in all periods of spermatozoa development, and plays a very important role in fish genetic breeding. We succeeded in creating an in vitro long-term culture method for *T. rubripes* SSCs, which can be stably passaged for more than 60 generations in vitro and display a variety of spermatogonial stem cell characteristics. The establishment of TrSSCs provides an important tool for the study of reproductive mechanisms in tiger puffer fish and other teleost.

## Figures and Tables

**Figure 1 animals-13-02959-f001:**
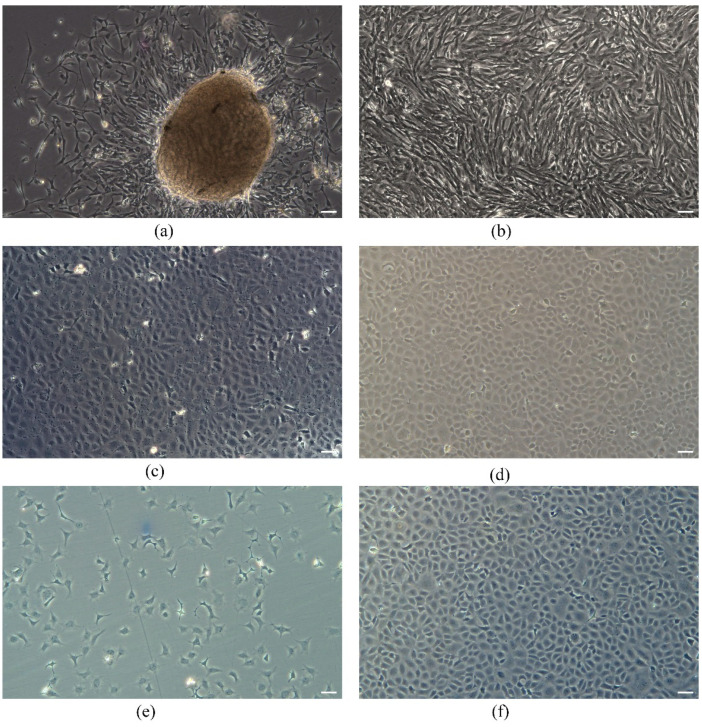
TrSSCs cultured in vitro. (**a**,**b**) Primary-cultured TrSSCs on the 5th and 20th days. (**c**,**d**) show subcultured TrSSCs in passages 15 and 35. (**e**,**f**) show the morphology of TrSSCs on the 2nd and 5th days of in vitro recovery. Scale bar = 50 μm.

**Figure 2 animals-13-02959-f002:**
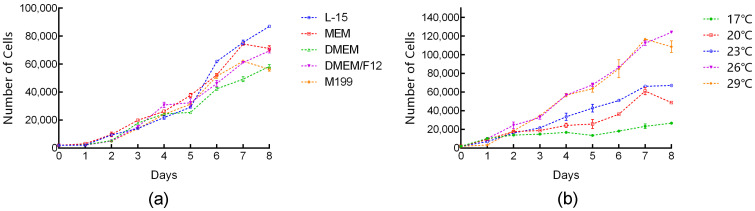
Factors influencing the growth rate of TrSSCs. (**a**) shows the growth rate of cells in different media containing 15% FBS at 23 °C. (**b**) shows the rate of cell growth at 17–29 °C in L-15 containing 15% FBS.

**Figure 3 animals-13-02959-f003:**
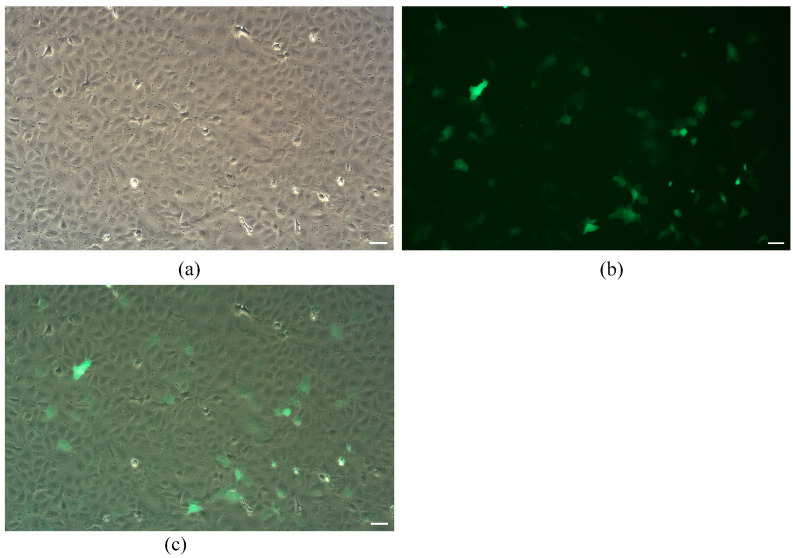
Transfection with the GFP reporter gene in TrSSCs. Bright-field (**a**), fluorescence (**b**) micrographs of passage 30 cells transfected with pEGFP-N1 and (**c**) merge image. Scale bar = 50 μm.

**Figure 4 animals-13-02959-f004:**
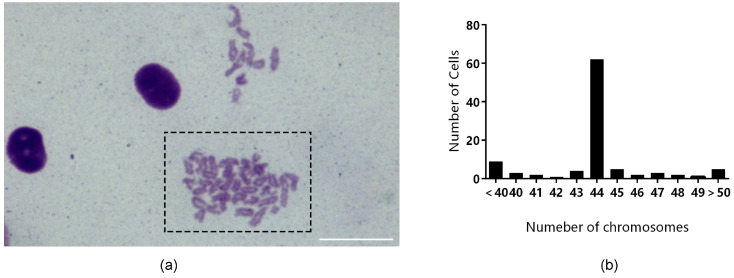
Chromosomal karyotype analysis of TrSSCs cells. (**a**) shows the chromosome morphology of TrSSCs at 40 generations. (**b**) shows the chromosome number distribution of 100 cells. Scale bar = 20 μm.

**Figure 5 animals-13-02959-f005:**
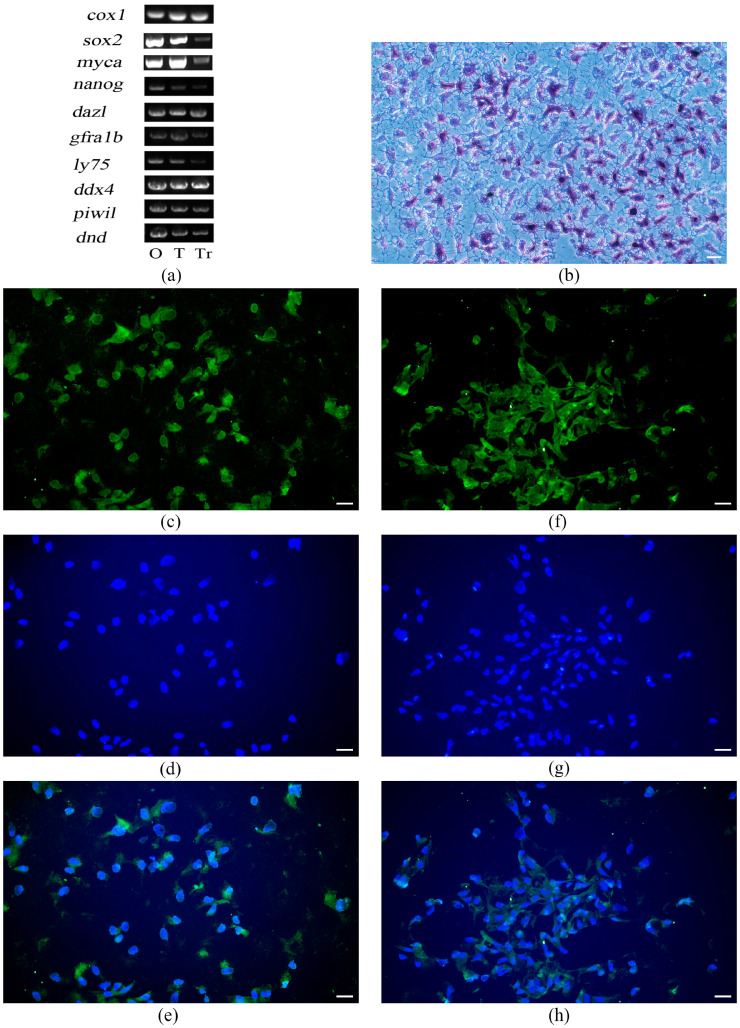
TrSSCs express spermatogonial stem cell-associated marker genes. (**a**) Expression of SSC-related genes and *cox1* in TrSSCs, testis, and ovary. (**b**) TrSSCs at passage 30 in alkaline phosphatase staining. (**c**) The localization of *ddx4* in TrSSCs at passage 30. (**d**) Cell nuclei were stained with DAPI. (**e**) Merge image. (**f**) The localization of *gfra1b* in TrSSCs at 30 generations. (**g**) Cell nuclei were stained with DAPI. (**h**) Merge image. Scale bar = 50 μm. Tr, TrSSCs; T, testis; O, ovary.

**Figure 6 animals-13-02959-f006:**
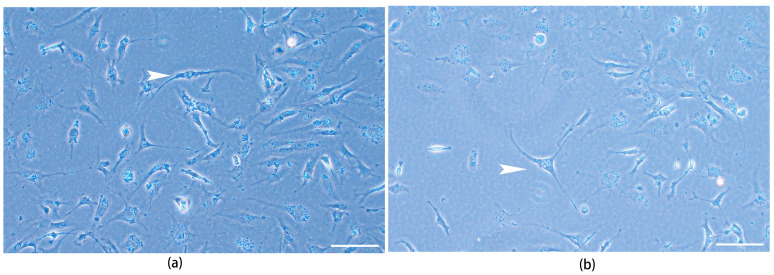
The types of terminally differentiated cells from TrSSC under retinoic acid treatment. The cell type indicated by the white arrowhead in (**a**) is a fibroblast, and the cell type indicated by the white arrowhead in (**b**) is an astrocyte. Scale bar = 50 μm.

**Table 1 animals-13-02959-t001:** Primers used for cloning and gene expression analysis.

Gene	F-Primer (5′ to 3′)
*cox1*-F	GAGGCTTTGGGAACTGATTA
*cox1*-R	GGTATTGTGAGATTGCTGGG
*ddx4*-F	AGAGGGATTAGACAAGGTGG
*ddx4*-R	GGCTCAAAACCCATATCCAG
*dnd*-F	TGATTCCTCTGTTCACCACT
*dnd*-R	GCACAAGTTATTGAGGAGCA
*piwil1*-F	TCAAACATACAGAGGAACGC
*piwil1*-R	GATTTCCTGACCTTTGTGCT
*gfra1b*-F	GAAGAAGGAGAAGAACTGCC
*gfra1b*-R	CATCACCGTACCAATGAGG
*ly75*-F	ACGTTACCCACTTTGACAAG
*ly75*-R	CATATCTAACTGGTGCAGGC
*nanog*-F	AAGTGACAGATTTGAGCAGC
*nanog*-R	GTAGTTTGGATGCCACTAGC
*dazl*-F	ACTATTTGGTCGGTATGGGT
*dazl*-R	TCCACAATCCACAAAGTTCC
*sox2*-F	AGGAAGATGGCGCAAGAAAA
*sox2*-R	GACATGTGTAGCCTGGTCTG
*myca*-F	GCGAGGACATCTGGAAGAAG
*myca*-R	AGACGTGGCATCTCTTCAAC
*vasa*	DIG-ACGATTCTCTTTAGATGGCATTCCAGGTGTA
*gfra1b*	DIG-GAAGGTGCCACAGACGGGATACAACGGA

## Data Availability

Data is contained within the article.

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
