# Peer review of "Establishment and Characterization of a Spermatogonial Stem Cell Line from Tiger Puffer Fish (Takifugu rubripes)"

_animals, 2023, doi:10.3390/ani13182959_

Round 1

Reviewer 1 Report

In this manuscript, the authors report the derivation and characterization of an SSC line from tiger puffer fish. This cell line enriched the fish SSC resource bank, and would be a useful tool for researches in germ cell differentiation. Here are some minor concerns.

1. the metaphase in fig 4 (a) is not typical, it is suggested to be replaced using the one with correct number. Additionally, the literature reporting the haploid chromosome number 44 of T. rubripes?

2. What method was used to calculate the transfection rate (40%) of the TrSSC? It is suggested to merge fig 3 (a) and (b) to show clearly the GFP positive cells.

3. the size of cox1 PCR product shown in fig 5(a) was in accordance with that of tiger puffer fish. How about the similarity in sequence alignement?

4. In alkaline phosphatase staining experiment, the cells with low confluence were suggested, in order to minimize the background color.

5. The capacity of differentiation of the the TrSSC in vitro or in vivo?

6. The cells were treated with colchicine for 4 h at 24 h?

thare are some typos.

Reviewer 2 Report

Invitro propagation of spermatogoonial stem cell is a way forward for fish reproduction. The author did a commendable job. However several concerns are there.

1. Several previous reports confirms that, SSC need to be selectively isolated with some specific markers, However author did the same without any marker. It is highly likely that, the author was able to raise a mixed population of SSC, gonia and somatic cells. Nothing has been clear from the manuscript. It is necessary to describe how the somatic cell and other non ssc cells were removed over the period. 

2. i believe author did 2D culture and the figure 1a showing colony formation of putative SSC. but non has been seen there after. why so?

3. logically speaking SSC remains in the body in low vasculature areas where cells might have restricted O2 supply. Authors data shows that L15 based mediums are better. saturated O2 culture might not be a good option for study?

4. Result section 3.5- SSC markers should be varified by stem cell marker like sox2, MYC etc. The markers used in this study are mostly relatively prevelent in gonia and subsequent stages.

5. Weak AP staining can be observed in differentiated cell, Figure 5a shows mild Nanog expression (a differentiation marker). the Figure 5b contains mixture of Cells, looks like contain strong AP staining others weak. better resolution picutre and compartive data should be provided.

6. the invivo settlement and differentiation potential should be explored.

NA
